# Leveraging Artificial Intelligence to Predict Health Belief Model and COVID-19 Vaccine Uptake Using Survey Text from US Nurses

**DOI:** 10.3390/bs14030217

**Published:** 2024-03-07

**Authors:** Samaneh Omranian, Alireza Khoddam, Celeste Campos-Castillo, Sajjad Fouladvand, Susan McRoy, Janet Rich-Edwards

**Affiliations:** 1Division of Women’s Health, Department of Medicine, Brigham and Women’s Hospital, Harvard Medical School, Boston, MA 02115, USA; ali.khoddam@northwestern.edu (A.K.); jr33@mgb.org (J.R.-E.); 2Department of Computer Science, University of Wisconsin-Milwaukee, Milwaukee, WI 53211, USA; mcroy@uwm.edu; 3Department of Media and Information, Michigan State University, East Lansing, MI 48824, USA; camposca@msu.edu; 4Stanford Center for Biomedical Informatics Research, Stanford University, Stanford, CA 94305, USA

**Keywords:** COVID-19 vaccination, healthcare providers, Nurses’ Health Study, vaccine hesitancy, health belief model, artificial intelligence, natural language processing, text classification

## Abstract

We investigated how artificial intelligence (AI) reveals factors shaping COVID-19 vaccine hesitancy among healthcare providers by examining their open-text comments. We conducted a longitudinal survey starting in Spring of 2020 with 38,788 current and former female nurses in three national cohorts to assess how the pandemic has affected their livelihood. In January and March–April 2021 surveys, participants were invited to contribute open-text comments and answer specific questions about COVID-19 vaccine uptake. A closed-ended question in the survey identified vaccine-hesitant (VH) participants who either had no intention or were unsure of receiving a COVID-19 vaccine. We collected 1970 comments from VH participants and trained two machine learning (ML) algorithms to identify behavioral factors related to VH. The first predictive model classified each comment into one of three health belief model (HBM) constructs (barriers, severity, and susceptibility) related to adopting disease prevention activities. The second predictive model used the words in January comments to predict the vaccine status of VH in March–April 2021; vaccine status was correctly predicted 89% of the time. Our results showed that 35% of VH participants cited barriers, 17% severity, and 7% susceptibility to receiving a COVID-19 vaccine. Out of the HBM constructs, the VH participants citing a barrier, such as allergic reactions and side effects, had the most associated change in vaccine status from VH to later receiving a vaccine.

## 1. Introduction

Healthcare providers worldwide became the frontline workers in battling COVID-19 by treating infected patients. Healthcare providers encountered numerous challenges increasing their risk of infection, such as inadequate Personal Protective Equipment (PPE), high workloads, and extended shifts [1]. In December 2020, US healthcare personnel began to be offered mRNA vaccines (Pfizer-BioNTech and Moderna) under emergency use authorizations (EUA) [2], a critical step toward protecting healthcare providers [3]. Several surveys conducted before the EUA documented that 8–18% of healthcare personnel expressed hesitancy about the safety and efficacy of the new vaccines [4]. Vaccine hesitancy dropped after the EUA and the vaccine rollout to healthcare personnel (HCP), yet a small minority of nurses remained vaccine hesitant even into Spring 2021 [5,6,7]. Determining why hesitancy lingered is a challenge. Heyerdahl et al. coined ‘unspoken vaccine hesitancy’ in describing when nurses may be uncomfortable expressing their vaccine-related concerns and beliefs due to the social and institutional pressure to get vaccinated [8]. 

Uncovering and understanding the factors behind vaccine hesitancy is crucial to increasing vaccine confidence in healthcare institutions and mitigating the spread of COVID-19. Previous studies have shown that open-ended questions in healthcare surveys are valuable resources to elicit respondents’ concerns [9] and can reduce response biases stemming from respondent beliefs about desired outcomes [10]. Therefore, to examine what underlies vaccine hesitancy among nurses and what factors influenced uptake, as expressed in their own words, we leveraged a national longitudinal survey of US nurses and applied machine learning (ML) methods to the open-text comments regarding vaccines in January 2021 (winter 2021 survey) as predictors of survey respondents’ vaccination status in March–April 2021 (Spring 2021 survey). Among a cohort of initially vaccine hesitant nurses who included comments, 40% received a COVID-19 vaccine by Spring 2021, but 60% remained unvaccinated. To help understand why, we employed constructs from the health belief model to help interpret the nurses’ rationales in their responses [11]. 

### 1.1. Gaining Insight from Open-Field Comments

Qualitative research can help understand the concerns and opinions of healthcare providers concerning COVID-19 vaccines. By analyzing respondents’ words, qualitative research enables us to gain insights into their beliefs, experiences, attitudes, behavior, and interactions [12]. Traditional qualitative analysis involves human coders reading respondents’ words and categorizing similar responses into multiple codes or themes through an iterative consensus process [13,14].

Manually analyzing high-volume, multi-class data collected during the qualitative research process is tedious and time-consuming. Over the past decades, ML has incorporated ideas from psychology, sociology, statistics, and mathematics to enable computers to predict outcomes based on specific predictors [15]. The core functionality of machine learning is to train computers to automatically solve problems like classification using data or prior experience [16]. We used ML in two ways: (1) assist with the qualitative research process by classifying text into multiple codes; (2) use the multiple codes to predict changes in vaccine status.

### 1.2. Health Belief Model Constructs

The codes we used to analyze open-field comments were developed from the health belief model (HBM), which can be applied to understand why healthcare providers receive or decline the COVID-19 vaccine. The HBM is a psychosocial model that researchers use to understand people’s health-related behavior, including their decision to adopt or decline disease prevention measures [11]. Previous research supports the utility of HBM constructs in predicting the uptake of vaccines and measures to prevent being infected with the virus that causes COVID-19 [17]. 

According to the HBM, whether an individual adopts a health-promoting behavior depends on five beliefs: (i) the individual’s perception of themselves as prone to disease or health risks (perceived susceptibility), (ii) their feeling of the severity of the disease (perceived severity), (iii) their challenge of taking preventative actions (perceived barriers), (iv) their view of the benefits of taking those actions (perceived benefits), and (v) their view of how well they can successfully implement the recommended health behavior (self-efficacy). Since VH individuals are hesitant about receiving a vaccine, we reasoned that the benefit and self-efficacy HBM constructs, which focus on the positives of vaccination, would not capture the health behavior of VH individuals [11]. It is recognized that individuals often weigh the cost–benefit ratio when deciding on vaccination. However, the diffusion of innovations theory suggests that the adoption of new healthcare technologies, like mRNA vaccines, follows an S-shaped curve, highlighting initial hesitancy due to unfamiliarity [18,19]. This framework provides context for the nuanced reassessment of classic cost–benefit ratio, differentiating initial vaccine acceptance from subsequent hesitancy phases [20]. Therefore, we focused on detecting severity, susceptibility, and barriers within the comments from nurses. 

## 2. Materials and Methods

We developed two ML models using open-text comments provided by VH nurses in two consecutive surveys in winter-Spring 2021. The first model identifies the presence of HBM constructs within the comments, and the second uses this information to predict change in VH between the two surveys.

### 2.1. Study Population and Data Collection

We first designed our study population to investigate the change in COVID-19 vaccine hesitancy among nurses who had participated in one of several ongoing large-scale studies. In the Spring of 2020, we launched a series of surveys regarding the COVID-19 exposures and experiences of participants in the Nurses’ Health Study II and Nurses’ Health Study 3. The Nurses’ Health Study II (NHSII) was initiated in 1989 with 116,429 female registered nurses (RNs) aged 25–42 residing in 14 states. The Nurses’ Health Study 3 (NHS3) is an open cohort launched in 2010 that continues to enroll nurses and nursing students aged 18 and older, born since 1 January 1965. This cohort includes RNs, licensed practical and vocational nurses, specialized RNs, and nursing students. The NHS3 cohort was expanded to include male nurses in 2015. Altogether, the NHS studies cover more than 280,000 participants.

In April–May 2020, we invited participants who had returned the most recent primary cohort questionnaires to complete a supplementary COVID-19 survey. Exclusions to the baseline invitation, such as the lack of a valid email address, are detailed in a previous study [1]. Of 105,662 invited participants, 58,606 (55%) completed the baseline survey. Respondents were surveyed again 1, 2, 3, 6, 9, and 12 months after the initial survey. Data collection for each survey was rolled out over three weeks. We restricted the current analysis to female credentialed current, active, and former nurses living in the United States who responded to the Spring 2021 survey [14]. This restriction left us with 38,788 participants.

We defined vaccine hesitant (VH) as a participant who answered ‘no’ or ‘unsure’ to the winter 2021 survey question, ‘Do you plan to receive a COVID-19 vaccine?’ According to our definition, of 38,788 participants, we excluded 18,395 participants who had already been vaccinated at the time the winter survey, 337 participants who had missing data on vaccination status, and 15,814 participants who indicated they had a plan to get a COVID-19 vaccine soon, leaving a population of 4242 VH individuals. The winter 2021 survey also provided two opportunities for respondents to express their thoughts: (1) an unprompted open-text box after the vaccine questions and (2) an open-text box at the end of the survey, prompting, ‘We are interested in learning more about your experiences during this pandemic. Please add anything else you would like to tell us here’. Of the 4242 VH individuals in winter 2021, we excluded 2741 participants who did not write any comments in the open-text boxes, leaving a study population of 1501 of VH participants with comments. Our analysis did not differentiate between participants based on prior COVID-19 infections, mirroring the initial U.S. vaccination policy which applied uniformly to previously infected and non-infected individuals. This approach was influenced by the early pandemic’s limited data on immunity from past infections and the urgent need for widespread immunity [21]. Figure 1 depicts the flowchart of our study population.

We used this dataset of 1501 comments as input to train an HBM ML model that predicts whether each comment belonged to any of the three HBM constructs (barriers, severity, and susceptibility) or was a non-HBM comment.

To monitor the vaccine status of survey participants three months after winter 2021, we asked the same question about vaccine status in the Spring 2021 survey. From the initial pool of 1501 VH individuals surveyed in winter 2021, 96 participants who did not specify their vaccine status were excluded, resulting in 1405 VH participants. Our focus for subsequent model development was narrowed down to these 1405 individuals who participated in both our winter and Spring 2021 surveys. As delineated in Figure 1, our second model is designed to predict the vaccine status of these VH individuals in Spring 2021.

### 2.2. Machine Learning Models

In this study, we trained two ML models: (1) a model to predict an HBM construct for each comment of VH participants in winter 2021 and (2) a model to predict a change in vaccine status of VH individuals from winter 2021 to Spring 2021. 

To develop the HBM model, four expert annotators created a training sample set by manually annotating 300 (16% of all VH comments) comments as one of the three mentioned HBM constructs or non-HBM-related comments. Table 1 shows the guidelines for classifying the three HBM constructs, definitions, and example comments from the winter 2021 survey. 

We utilized the Python programming language and its built-in scikit-learn free software machine learning library for developing and analyzing our ML models. Before training our two models, we first preprocessed the data. This involved data cleaning, segmenting each comment into words, eliminating low-variance terms (words) related to the output variables and converting words to numerical values. The details of the data preprocessing and feature selection can be found in Appendix A.

For each training task, we first adopted an 80/20 train-test split to initially partition the train set. We then trained five ML models: random forest (RF) [22], multinomial naïve Bayes (Multinomial NB), logistic regression, scholastic gradient descent (SGD) [23], and a multi-layer perceptron neural network (NN) [24], and then selected the model with the best performance. Furthermore, to evaluate the robustness and generalizability of the ML models, we employed K-fold cross-validation with K set to 10 for the training set of the ML models. The 10-fold cross validation approach allowed us to utilize every comment for both training and validation in different folds. The details of the data preprocessing and the features used in the ML models can be found in Appendix A.

## 3. Results

Our analysis of input feature reduction dimensionality by the variance threshold method revealed that a subset of 705 high-variance input features is the best for training an HBM model for comments related to COVID-19 vaccination, and a subset of 430 input features is the optimal size for training a change in the vaccine hesitancy model. To provide a more comprehensive understanding of both models performance, we used weighted average recall (sensitivity), precision, F1-measure, accuracy, and area under the curve for the receiver operating characteristic curve (AUC-ROC).

Recall, also known as sensitivity and true positive rate, is a metric that measures the proportion of accurate positive predictions among all possible positive predictions. Recall ranges from 0 to 1 and high recall indicates that the model can identify most of the positive instances and is calculated using the equation below.
$$recall=\frac{True\;Positive}{True\;Positive+False\;Negative}$$

We explain the details of each model in the following sections. A precision measure calculates how many correctly positive predictions were made and is calculated using the equation below: $$precision=\frac{True\;Positive}{True\;Positive+False\;Positive}$$

F1-measure is a harmonic mean of precision and recall and ranges from 0 (prediction failure) to 1 (perfect prediction). A high F-measure indicates both good precision and recall, meaning the model has a good balance between minimizing false positives and false negatives.
$$\text{F1-measure}=2*\frac{Precision*Recall}{Precision+Recall}$$

Accuracy is a common metric used to evaluate the performance of a classification model. It measures the proportion of correctly classified instances out of the total instances in the dataset.
$$accuracy=\frac{True\;Positive+True\;Negative}{True\;Positive+False\;Positive+False\;Negative+True\;Negative}$$

The area under the curve (AUC) of the ROC curve is a scalar value that quantifies the overall performance of a classification model. A model with no predictive power would have an AUC of 0.5. This is equivalent to a random guess. A model with perfect predictive power would have an AUC of 1. A steeper ROC curve generally indicates better model performance.

### 3.1. HBM Prediction

Table 2 depicts the performance of the models we trained to predict the label for the unlabeled dataset (the rest of data that we did not use for HBM annotation). The label was any of the three constructs of the HBM (barrier, susceptibility, and severity) or a non-HBM-related construct. The NN model outperformed other models with 82% accuracy. As Figure 7b shows the NN model demonstrated robust performance with an area under the ROC curve (AUC-ROC) of 91%, indicates its high discriminatory ability. Figure 2, Figure 3, Figure 4 and Figure 5 show the most influential meaningful words in detecting HBM constructs in all comments from VH individuals. We used the standard deviation to measure the spread of words in the comments for each category. As shown in Figure 4, due to a low number of comments in this category and uniform appearance of words in the comment, standard deviation is also uniform. The apparent uniformity in the top 30 words in our findings aligns with established patterns in health communication, particularly in vaccine hesitancy contexts. Communication strategies in public health often utilize a consistent set of terms to address specific health behaviors and perceptions [25]. This consistency in language use, especially in discussions surrounding susceptibility to diseases, suggests a standardized approach in health messaging, leading to similar levels of word effectiveness across various informational sources. 

The results of our subsequent data analysis of the words in VH comments for each HBM construct are shown in Figure 6. We found that 35% of the comments from the VH nurses indicated perceived barriers to vaccination and that this was the most frequently expressed belief type. Our ML analysis on HBM constructs also suggested more specific details about these barriers; the most significant barriers to receiving the vaccine for the VH participants appear to be side effects and allergic reactions, with other concerns regarding pregnancy and fertility that were potentially related to the lack of research on the vaccine side effects within these domains.

The HBM classification also revealed that 17% of VH comments mentioned a perceived lack of severity of COVID-19. The most related topics under this category tended to mention high recovery rates from contracting COVID-19. An additional 7% of VH comments mentioned low susceptibility to contracting COVID-19. This category mentioned words suggesting respondents felt they already had a robust immune system, such as ‘supplements,’ ‘boost,’ and ‘diet.’

### 3.2. Results for Vaccine Status Prediction

The model used the comments from VH individuals in winter 2021 to predict vaccine status of the VH in Spring 2021. After expressing vaccine hesitancy in the winter 2021 survey, 40% of nurses received a COVID-19 vaccine by Spring 2021, while 60% remained unvaccinated. Table 3 shows the performance of the trained models for predicting the vaccine status from VH comments. The best accuracy achieved was 89% using a NN. Figure 7a shows the AUC-ROC curve of the model performance.

## 4. Discussion

This study used open-text comments collected in winter 2021 to predict vaccine status and hesitancy three months later. Throughout this time, the vaccines were accessible only under an emergency use authorization and it was not yet mandated for nurses to be vaccinated. The winter 2021 survey results showed that 11% of the participating nurses expressed vaccine hesitancy and 4% wrote at least one comment. In Spring 2021, 60% of the VH in winter 2021 remained hesitant three months later. Our results suggest that while most early VH may be due to perceived barriers, there is a small cohort for whom a belief in low COVID-19 severity or low personal susceptibility will linger. Our findings extend previous studies using qualitative coding to analyze COVID-19-related surveys [26,27,28]. 

We found the most commonly occurring HBM construct within the comments was perceived barriers, suggesting concerns over the tangible and intangible costs associated with getting the vaccine were prevalent among VH nurses. Specific costs mentioned included side effects and allergic reactions, indicating opportunities to develop messaging to allay these concerns. Pregnancy and fertility concerns were also frequently mentioned among comments assigned to this construct, likely owing to the lack of evidence regarding the vaccine’s safety among pregnant women and the messaging uncertainty among obstetricians [29]. 

This study takes the unique approach of using ML to categorize open-text comments into HBM constructs. Both the HBM classifier’s 82% accuracy rate to predict HBM constructs and 89% accuracy rate for predicting vaccine status from text reveal that implementing ML has the potential to automate the qualitative research project. In addition, the area under the ROC curve for both models exceeding 80% indicates robust model performance, signifying the models’ effectiveness in predicting the targets compared to random guessing. 

Our methods would also be helpful for future research augmenting ML with health behavior models. Such theoretical models of human behavior boast clear demarcations between constructs, necessitating corresponding ML methods to identify distinguishing features (e.g., words). Our analysis of feature selection revealed that a subset of high-variance words from each comment, rather than the entire comment, leads to improved performance in machine learning models. Additionally, we observed that excluding words representing two classes equally enhanced model performance. Similar approaches will likely yield better performance of ML methods that adopt a theoretical framework.

In our study, among several classical ML models we tested, the neural network model surpassed all other models, achieving the highest performance in both the HBM model and vaccine status prediction model. We also discovered that applying a variance threshold for input feature dimensionality reduction helped to optimize ML model training.

In our model, keywords such as ‘efficacy’ and ‘mutate’ were identified as relevant, which presents an intriguing aspect of our findings. Previous studies have shown that public understanding and perceptions of vaccine efficacy and viral mutation play a significant role in vaccine uptake. For instance, it is emphasized the impact of media narratives on public perception, where terms like ‘mutate’ can evoke fear or skepticism about vaccine adaptability and long-term effectiveness [30]. Thus, the relevance of these terms in our model aligns with the broader discourse in health communication research. This paradoxical aspect of our findings, where awareness of scientific concepts may inadvertently contribute to hesitancy, underscores the complexity of vaccine communication strategies. It suggests a need for health authorities to engage in more nuanced messaging that addresses specific concerns and misconceptions, a point echoed in the literature [31]. 

Public questioning of vaccine efficacy may have been tied to perceptions that the vaccine was of only marginal (or no) benefit to people who had already achieved some degree of immunity due to infection. The U.S. public health approach did not initially differentiate vaccine recommendations based on prior infection, reflecting a complex interplay of factors including operational challenges and the evolving understanding of the virus and immunity. In the U.S., the priority was to expedite the vaccination process to achieve broad immunity, particularly in light of emerging variants. This approach, while contentious, was influenced by a need for a clear and unified public health strategy amidst a rapidly developing situation. 

Readers should bear in mind the study’s limitations when drawing conclusions. First, our findings only apply to English-speaking nurses in the US, as only English text was used to develop the model. Additionally, the cohort primarily consisted of individuals of white ethnicity; hence, our ML model has not been trained to recognize how other racial groups feel about the COVID-19 vaccines. Moreover, not every VH nurse responded to the open-text questions used in the analysis. Lastly, in considering susceptibility, this category could potentially carry a valence. For instance, a comment could mention that they are choosing to not receive a vaccine because they are not susceptible to COVID. Opposingly, a comment could mention that they are choosing to receive a vaccine because they are susceptible to COVID. Since our study population is VH individuals, we chose to focus our analysis on the former valency of the susceptibility category. By expanding our study population to all individuals, future studies can consider both valences. Nonetheless, we were able to obtain a high accuracy rate. 

Our approach underscores the emergence of non-HBM constructs as significant predictors of VH. Research has consistently highlighted diverse factors influencing VH, including perceived risks and benefits, social norms, and trust in health authorities, which extend beyond traditional HBM constructs [32,33]. Our study’s identification of 40% non-HBM constructs in analyzing VH among nurses parallels these findings, demonstrating the efficacy of our methods in capturing a broad spectrum of determinants affecting health behavior decisions. The revelation of non-HBM themes, such as trust in vaccine efficacy and healthcare systems, mirrors the concerns and motivations identified in populations beyond healthcare professionals, suggesting a universal applicability of these insights. For instance, Xiao and Wong emphasized the role of perceived behavioral control in vaccination intentions, a concept resonating with the non-HBM codes discovered in our analysis [34]. This finding highlights the need for an automated process which can provide insight into health-related decisions. With the use of NLP, we can capture context-specific factors across demographics which traditional HBM might not capture.

Our study’s methodology, particularly in leveraging a neural network for analyzing vaccine hesitancy, offers a template for future researchers tackling emergent public health crises. The adaptability of our model means it can be retrained with relevant data to explore attitudes in different populations, not limited to nurses, providing valuable insights into public sentiment in real-time. This capability is especially pertinent for rapid response scenarios, where understanding public perception is crucial for effective health communication and policymaking. 

## Figures and Tables

**Figure 1 behavsci-14-00217-f001:**
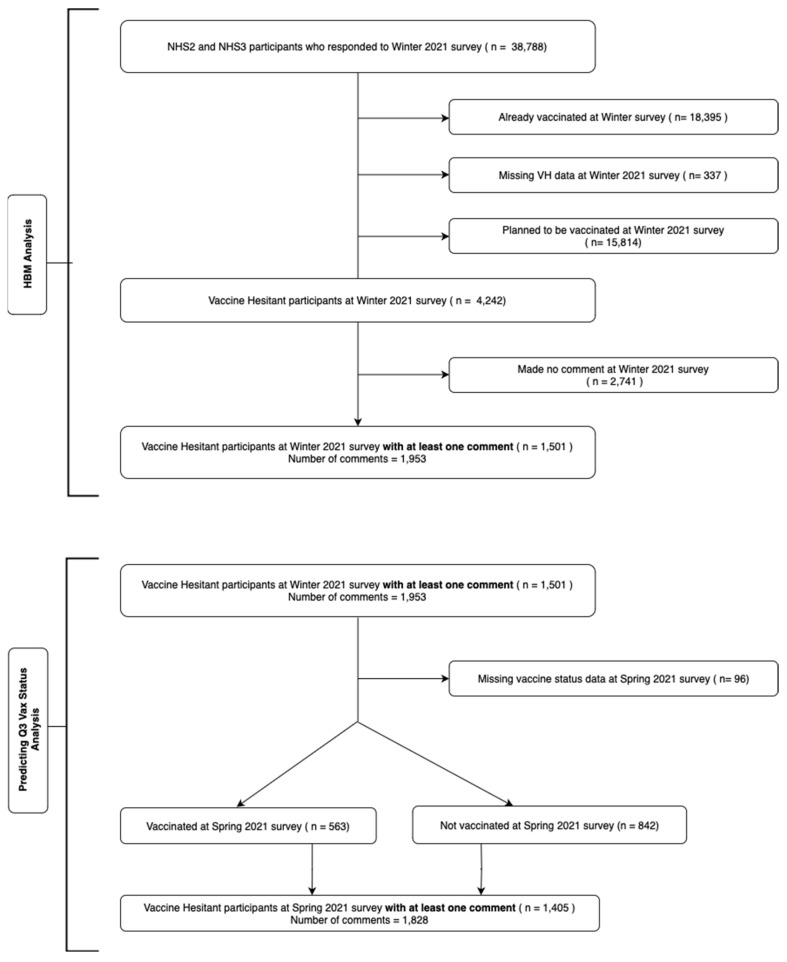
The flowchart diagram of the population lection for training ML models. Model 1 categorizes comments from winter 2021 into HBM constructs. Model 2 uses comments from VH individuals in winter 2021 to predict vaccine status in Spring 2021.

**Figure 2 behavsci-14-00217-f002:**
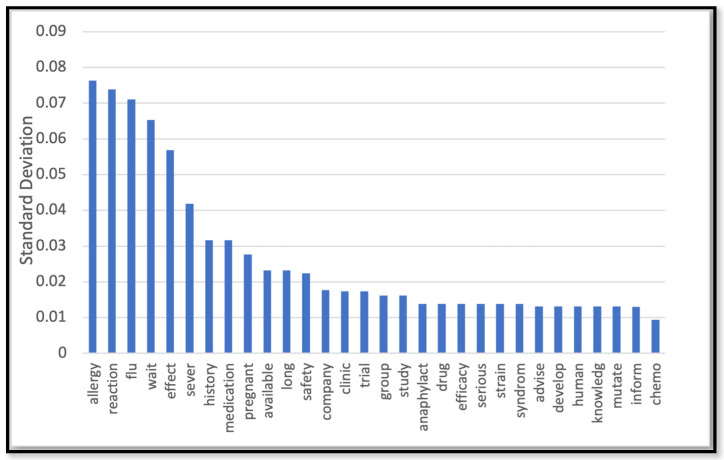
Top 30 most effective words in detecting the barrier construct.

**Figure 3 behavsci-14-00217-f003:**
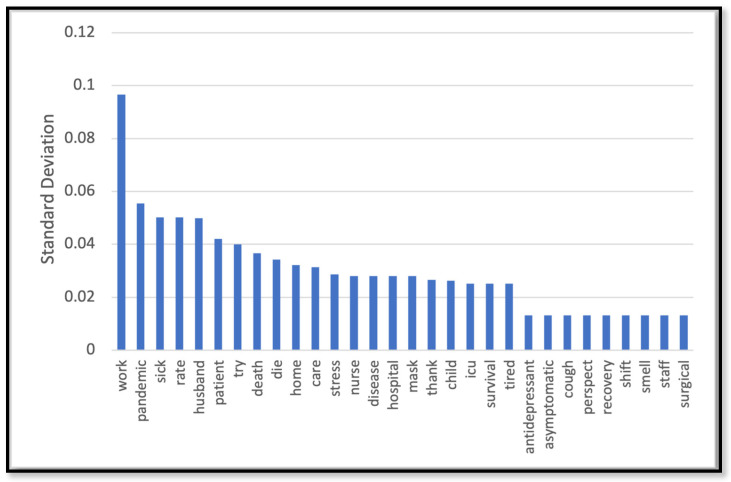
Top 30 most effective words in detecting the severity construct.

**Figure 4 behavsci-14-00217-f004:**
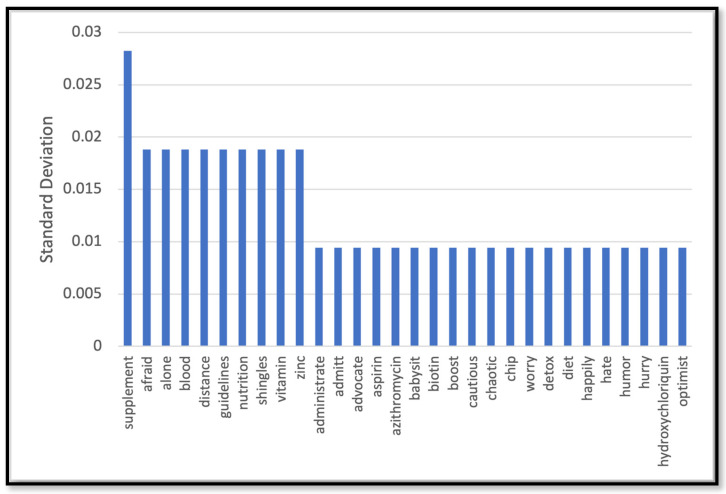
Top 30 most effective words in detecting the susceptibly construct.

**Figure 5 behavsci-14-00217-f005:**
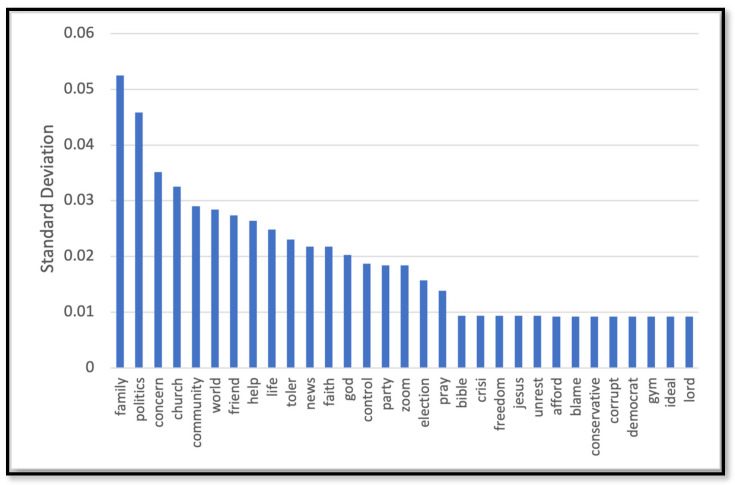
Top 30 most effective words in detecting the non-HBM group.

**Figure 6 behavsci-14-00217-f006:**
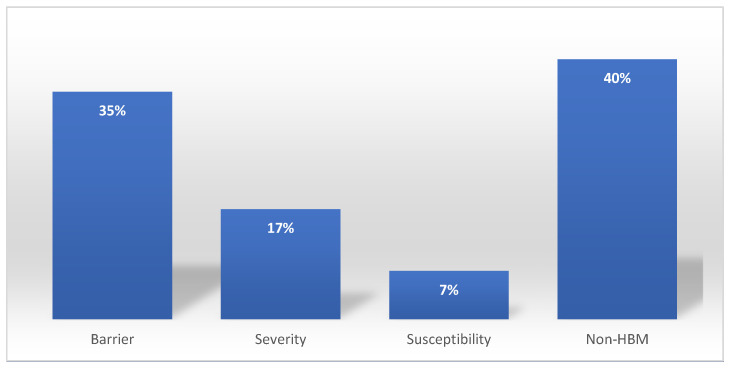
Distribution of the health belief model (HBM) classification on 2424 comments of nurses on the winter 2021 survey.

**Figure 7 behavsci-14-00217-f007:**
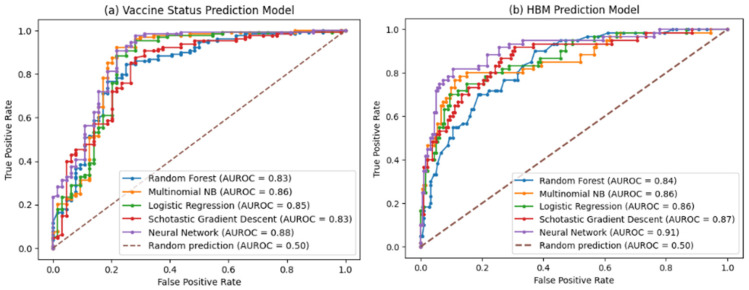
(**a**) Shows the AUC-ROC curve of the vaccine status prediction model; the neural network model outperformed other models with the AUC of 0.88. (**b**) shows the AUC-ROC curve of predicting HBM constructs. As shown the neural network model achieved the highest AUC = 0.91 among other models.

**Table 1 behavsci-14-00217-t001:** Health belief model constructs, definitions, and example comments from the VH comments in the winter 2021 survey.

HBM Construct	Definition	Example Comment
**Perceived** **Barriers**	Belief about the tangible and psychological costs of the advised action.	“Not fully tested or approved and is unnecessary for someone low risk like me. Also has not been studied for its effects on fertility and future pregnancies. I also know someone person that died 2 days after receiving the vaccine who had no medical conditions other than being overweight”.
**Perceived** **Severity**	Feelings about the seriousness of contracting an illness or of leaving it untreated include evaluations of both medical and clinical consequences (for example, death, disability, and pain) and possible social consequences (such as the effects of the conditions on work, family life, and social relations).	“There is a 99.7% survival rate for someone my age anyway”.
**Perceived** **Susceptibility**	Belief about the chances of getting a condition or disease.	“I honestly don’t know what to believe anymore. I consider myself healthy for my age; no co-morbidities; and it take vitamins and supplements that have been proven to boost the immune system. So I see no reason to take a rushed vaccine. I have never gotten a flu shot and have never gotten the flu either”
**Non-HBM**	Comments that do not fall into any of the above categories.	“I needed to travel in mid January as my dad had major surgery and needed someone to be with him”.

**Table 2 behavsci-14-00217-t002:** Machine learning performance results for the HBM prediction model using different combinations of vectorization methods (column) and classifiers (rows).

Machine Learning Algorithm	Recall (Sensitivity)	Precision	F1-Score	Accuracy	AUC-ROC
Random Forest	0.60	0.61	0.59	0.60	0.84
Multinomial NB	0.60	0.57	0.56	0.72	0.86
Logistic Regression	0.58	0.54	0.55	0.68	0.86
SGD	0.61	0.63	0.61	0.67	0.87
Neural network	0.82	0.85	0.79	0.82	0.91

**Table 3 behavsci-14-00217-t003:** Machine learning accuracy results for the vaccine status prediction model using different combinations of vectorization methods (column) and classifiers (rows).

Machine Learning Algorithm	Recall (Sensitivity)	Precision	F1-Score	Accuracy	AUC-ROC
Random Forest	0.71	0.76	0.72	0.78	0.83
Multinomial NB	0.84	0.89	0.86	0.88	0.86
Logistic Regression	0.69	0.86	0.71	0.79	0.85
SGD	0.78	0.81	0.79	0.82	0.83
Neural network	0.89	0.90	0.89	0.89	0.88

## Data Availability

Our public website (www.nurseshealthstudy.org, accessed on 4 December 2023) includes a brief description of the Nurses’ Health Study cohorts, all questionnaires, and a description of resource sharing procedures. An automated online form requests the applicant to briefly describe the hypothesis and aims, variables needed, etc. Requests are presented to the cohort investigator meetings every other week, and replies are provided within 2 weeks. After appropriate institutional IRB approvals, data access occurs in one of three ways: (1) the external investigator, with a password-protected login to our system, securely accesses and analyzes cohort data/specimen results; (2) the external investigator requests collaboration with or support of an internal investigator and/or programmer who conducts the analyses on the external investigator’s behalf; or (3) a specific dataset and data dictionary are created to send to the external collaborator. Most often, investigators are provided with direct access to the cohort computing system; the third option is usually reserved for consortia projects pooling data from cohorts. Access is provided with evidence of human subjects training (required by our IRB) and a standard data use agreement. Login to the computing system is easily done from anywhere in the world, with a password-protected logon, which provides access to data systems and intranet site, with educational materials and documentation. These are the same online data, materials, and tools accessed by internal investigators.

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
