# Peer review of "Leveraging Artificial Intelligence to Predict Health Belief Model and COVID-19 Vaccine Uptake Using Survey Text from US Nurses"

_behavsci, 2024, doi:10.3390/bs14030217_

Round 1

Reviewer 1 Report

Comments and Suggestions for Authors

The paper addresses a critical topic that might be a crucial step in combating future pandemics. In this study, the authors performed multiple surveys and gathered a very valuable dataset. In addition, the authors compared multiple ML models to characterize the comments made by nurses to provide a better understanding of the barriers that have led to vaccine hesitancy.  Finally, another set of models has been trained to predict the vaccination status of hesitant nurses. The described methodology shows another way in which AI and ML can be used to extract meaningful information from the data in future pandemics.

However, several aspects need to be addressed to improve the manuscript:

·         Only 300 (16%) posts were labeled by annotators for the training. The poor performances of the models can be due to the very small training sample size. Additional data labeling is highly recommended to achieve F1 scores higher than 0.9.

·         It is not clear how the authors performed the train-test splits. Usually, researchers use 80% of the labeled dataset for training and 20% for testing. Considering the already low labeled dataset size (300 comments), this would be a pretty small number of comments used for training. Finally, a very small sample of comments used for testing could provide misleading results when it comes to models’ performances.

·         The authors report the average F1, precision, recall, accuracy, and AUC of each machine learning model trained. However, it is not clear how many times each model ran. For example, are these the average values across 100 random test-train splits? Did the authors include validation in the models? This has to be clarified in the manuscript. As of now, it sounds like the authors only trained each model a single time, which might not provide a complete picture of how each model performs.

·         Therefore, based on the points mentioned above, the results might contain a certain bias.

·         It would be interesting to discuss more on the results presented in the manuscript. For example, authors report the most effective words found by their models for detecting severity construct, susceptibly construct, etc. (figures 2-5). Did the authors find any of these surprising?

·         It is also not clear how this methodology can be used by researchers in the future. It could be useful if authors dedicate a couple of sentences explaining how the models can be leveraged in future emergencies and how they can be applied to the general population besides nurses. 

Comments on the Quality of English Language

Minor edits 

Reviewer 2 Report

Comments and Suggestions for Authors

The research idea shows promise and demonstrates a degree of innovation. However, to fully capitalize on its potential, there is room for improvement in the actual implementation, including some shortcomings that combined warrant major revision.

1) Did you perform any evaluation on the test set of your neural network? The argument that a neural network, not tested for overfitting, was providing very high predictive value on data that it had been extensively trained is less than impressive. For validation purposes, at the very least, you should split your dataset, train it on one subset, and test it on another. As the intent of your work is to "leverage artificial intelligence," you really should leverage it correctly.

2) “Figure 4. Top 30 most effective words in detecting the susceptibly construct.”

Could you please review this graph? Assuming that you are correct, could you explain, possibly in your article, why those values appear to be so uniform?

3) It's possible that I might be overlooking this information, but are you controlling for whether a person has already been infected? I'm aware that the USA stood out as a country that, for mysterious reasons, did not fully acknowledge the otherwise seemingly uncontroversial fact that antibodies from past infections tend to provide some protection afterward. I mean, if you compare US policy with, let's say, the EU, one was generally considered not requiring vaccination for half a year if they had recovered from the infection. If you consider subsequent studies, they indicated that hundreds of people with a prior infection needed to be vaccinated to prevent a single subsequent case, raising doubts about the medical benefit for the patient with vaccines with a rather high rate of side effects.

As your article is intended for a global audience, you should address this issue of US uniqueness. Did you count as vaccine-hesitant people who did not follow policy, which was highly contentious from its inception and especially with the benefit of hindsight appears as dubious from a medical standpoint? Or maybe you implicitly dropped such people from the analysis? Certainly, you could simply consider local regulations as the benchmark for the delineation of vaccine hesitancy, but as your article is intended for a global audience, you should provide an explanation about it, either in limitations or your methodology.

Just in case you are uncertain about the very limited gains from the vaccination of people who acquired natural immunity:

doi.org/10.1016/ S1473-3099(22)00143-8

4) >Perceived Severity - “There is a 99.7% survival rate for someone my age anyway.”

>Perceived Susceptibility  - “95-98% cure rate I am (in) good health.”

At first, I wondered whether it's me who is unable to detect nuance between those two sentences. Later, I decided to look up the original paper which you use for the selection of the theoretical model: 

“Perceived Susceptibility

Individuals were believed to vary widely in their acceptance of personal susceptibility to a condition. At one extreme might be the individual who denies any possibility of his contracting a given condition. In a more moderate position is the person who may admit to the “statistical” possibility of a disease occurrence, but a possibility that is not likely to happen.”

doi.org/10.1177/109019817400200403

So, this selection is clearly done in a way contradicting the theory. Low perceived susceptibility would mean that a person considers the chances of contracting an infection as very low, perhaps due to following safety precautions or working from home. Please not only correct this sentence, but also check whether otherwise the selection was done correctly. 

5.1) While looking at words that your model I wondered to what extend you performed here hypothesis-driven analysis, where some observations that did not fit theory were simply neglected.

For example:

>“Since VH individuals are hesitant about receiving a vaccine, we reasoned 95 that the benefit and self-efficacy HBM constructs, which focus on the positives of vaccina-96 tion, would not capture the health behavior of VH individuals [11].”

I'd expect people to strive to estimate a cost-benefit ratio and be more likely to tolerate side effects or foregoing standard medical product safety test procedures in the case of benefits being impressive, while the same might not be said when benefits become much smaller. This process is well-documented, even in relation to COVID-19 vaccines, as mentioned in studies showing "booster hesitancy." Individuals who enthusiastically queued for the primary course, expecting high and lasting protection against the deadly infection, somehow became vaccine-hesitant and were not interested in subsequent doses that provided them with a few months of elevated protection against symptomatic disease.

Look up:

https://doi.org/10.3390/vaccines9121424

https://doi.org/10.3390/vaccines9121437

(The second paper explicitly states that those healthcare workers openly stated that the cost-benefit ratio is unfavorable, so you may find it useful for your analysis.)

5.2) Setting this technical aspect aside, where you might have your own interpretations, your model detects key words such as "efficacy" or "mutate" as relevant. While HARKing would be undesired practice and likely the impact is weak enough that splitting this variable would be a bad pick modeling-wise, I'd strongly recommend elaborating on the issue in the discussion section. Such paradoxical findings are usually highly valuable.

5.3) Additionally, while you attempt to analyze using the Health Belief Model and reasonably consider the possibility that some unrelated mechanisms could be at play, you discover that "Non-HBM" factors are responsible for 40% of the answers. Moreover, based on your keywords, there appears to be U.S.-specific sociopolitical dynamics where certain forms of medical treatment could become a political issue. This issue also would require some discussion. 

6)I also have some concerns regarding your limitations. Firstly, in such models, there is an implicit assumption that people make decisions based on what is intended to be a logical analysis of available information, and you neglect the issue that behaviors in such a new situation were strongly shaped by social proof. Effectively, a vaccine-hesitant person, when facing a situation that is being deemed as ambiguous, is assessing the default reaction of their peers and is likely to mimic what is normal behavior within the group. This paradoxically should be deemed as both a limitation of your model and also as a ceiling for the expected predictive value of any model that would capture a snapshot of individual attitudes at a particular moment. As any way of assessing their attitude and beliefs would not detect any subsequent subtle social pressure. For evidence of such social proof behaviors in relation to COVID-19 vaccination:

doi.org/10.3390/vaccines10040528

>“there is a small cohort for whom a belief 279 in low COVID-19 severity or low personal susceptibility will linger.”

You should mention here as limitations that you have to model it as a "belief," while studies on late vaccination indicate that instead of having something akin to a "belief," vaccine-hesitant people are rather displaying some kind of risk assessment of the dynamics situation. Effectively, when there is a mounting infection wave, there is also a detectable surge in vaccinations.

doi.org/10.3390/vaccines10091523

You are mentioning in limitations language or race issues, while overlooking elephant in the room – politics. If one tried to repeat this study in my country, I’d not expect staggering differences due to some ethnic or linguistic factors, while I’d expect serious divergence on sociopolitical dynamics. 

Round 2

Reviewer 1 Report

Comments and Suggestions for Authors

All suggestions/comments have been addressed.

Author Response

Dear Reviewer,

I hope this message finds you well. We would like to express our sincere gratitude for the time and effort you have dedicated to reviewing our manuscript titled "Leveraging Artificial Intelligence to Predict Health Belief Model and COVID-19 Vaccine Uptake Using Survey Text from US Nurses" with the ID [behavsci-2784657]. We are truly appreciative of your positive feedback and are pleased to learn that all the suggestions and comments have been satisfactorily addressed.

It has been an enriching experience, and we have endeavored to ensure that the manuscript now accurately reflects the high standards expected by MDPI Behavioral Sciences Journal.

Thank you once again for your invaluable support and feedback.

Warm regards,

Authors.

Reviewer 2 Report

Comments and Suggestions for Authors

Good work!

As a side note, while your response regarding why Figure 4 may give the wrong impression was somewhat diplomatic, I believe I've identified the issue. That specific category has the lowest number of responses. Consequently, there are categories with equal standard deviation because they contain only 3, 2, or 1 response(s). So yes, you plotted it correctly; nevertheless, anyone carefully scrutinizing this graph may begin to raise eyebrows. You may consider hinting such explanation there.

Author Response

Dear Reviewer,

We would like to express our deepest gratitude for the constructive feedback and the positive remarks received from you regarding our manuscript titled "Leveraging Artificial Intelligence to Predict Health Belief Model and COVID-19 Vaccine Uptake Using Survey Text from US Nurses". We are particularly appreciative of your side note concerning Figure 4 and have incorporated your suggestion into our manuscript. Your observation is invaluable, and we acknowledge the importance of providing a clearer explanation to ensure that readers do not misinterpret the graph presented. 

Once again, we are grateful for the opportunity to refine our work through this review process and for the constructive guidance we received. It has been an enriching experience, and we have endeavored to ensure that the manuscript now accurately reflects the high standards expected by MDPI Behavioral Sciences Journal.

Warm Regards,

Authors.

____________________________________________

point-by-point response to the reviewer’s comments

Comments and Suggestions for Authors

Good work!

As a side note, while your response regarding why Figure 4 may give the wrong impression was somewhat diplomatic, I believe I've identified the issue. That specific category has the lowest number of responses. Consequently, there are categories with equal standard deviation because they contain only 3, 2, or 1 response(s). So yes, you plotted it correctly; nevertheless, anyone carefully scrutinizing this graph may begin to raise eyebrows. You may consider hinting such explanation there.

Response: Thank you for your insightful side note. As you accurately mentioned, due to the small number of responses for the susceptibility category of HBM constructs, the standard deviations of the words are equal. We have included this note in our manuscript on page 8 as follows. Thank you once again for the thoughtful and detailed review process, which has significantly contributed to the improvement of our manuscript.

updated text in the manuscript

Figures 2-5 show the most influential meaningful words in detecting HBM constructs in all comments from VH individuals. We used the standard deviation to measure the spread of words in the comments for each category. As shown in Figure 4, due to low number of comments in this category and uniform appearance of words in the comment, standard deviation is also uniform. The apparent uniformity in the top 30 words in our findings aligns with established patterns in health communication, particularly in vaccine hesitancy contexts. Communication strategies in public health often utilize a consistent set of terms to address specific health behaviors and perceptions [25]. This consistency in language use, especially in discussions surrounding susceptibility to diseases, suggests a standardized approach in health messaging, leading to similar levels of word effectiveness across various informational sources.